# High Antennal Expression of *CYP6K1* and *CYP4V2* Participate in the Recognition of Alarm Pheromones by *Solenopsis invicta* Buren

**DOI:** 10.3390/insects16010043

**Published:** 2025-01-05

**Authors:** Xinyi Jiang, Jiacheng Shen, Peng Lin, Youming Hou

**Affiliations:** State Key Laboratory of Ecological Pest Control for Fujian and Taiwan Crops, Fujian Agriculture and Forestry University, Fuzhou 350002, China; xinyijiang1116@163.com (X.J.); jc_shen_cnsh@outlook.com (J.S.); m15959689635@163.com (P.L.)

**Keywords:** red imported fire ant, alarm pheromone, cytochrome P450s, RNA interference, electroantennography

## Abstract

Studies have shown that cytochrome P450s (CYPs) in the antennae of insects are involved in the entire process of olfactory recognition of odor compounds. In this study, through transcriptome technology and RT-qPCR, we identified CYPs that are specifically and highly expressed in the antennae of *Solenopsis invicta* worker ant (*SinvCYP6K1* and *SinvCYP4V2*). Results from RNA interference (RNAi) combined with electroantennogram (EAG) and behavioral experiments demonstrated that *SinvCYP6K1* and *SinvCYP4V2* are involved in the recognition process of 2-ethyl-3,6(5)-dimethylpyrazine by *S. invicta* worker ant.

## 1. Introduction

Insects possess highly sensitive and well-developed olfactory systems. The diverse chemical receptors that insects possess assist them in recognizing a variety of odor molecules, including host plant volatiles and pheromones. These molecules play an integral role in insects’ search for food, mates, oviposition sites, and protection from predators [1,2]. The olfactory proteins that are typically involved in the process of insects receiving odor signals include odorant binding proteins (OBPs), chemosensory proteins (CSPs), odorant receptors (ORs), ionotropic receptors (IRs), gustatory receptors (GRs), sensory neuron membrane proteins (SNMPs), and odorant-degrading esterases (ODEs) [3,4]. Chemical molecules in the environment gain access to the antennal sensillum lymph via the micropores on the antennae, bind to OBPs, and are transported via the lymph to the ORs located on the dendritic membrane. Once the ORs are stimulated, the chemical signal is converted into an electrical signal that is transmitted to the central nervous system, thereby triggering the corresponding behavioral response in the insect. Following the interaction of the ORs, the signaling molecules must be deactivated, a process that is primarily mediated by ODEs [5]. Once chemical signals have been transmitted to the central nervous system, it is imperative that they are deactivated without delay to avert any potential damage to the nervous that could result from the continuous stimulation of chemical molecules [4]. ODEs have the capacity to inhibit the aggregation of odor molecules within the olfactory receptor cells. They are primarily involved in the metabolic processes of exogenous compounds, pheromones, and various allelochemicals. ODEs include antenna-specific or antenna-enriched cytochrome P450s (CYPs), glutathione S-transferases (GSTs), carboxylesterases (CXEs), uridine diphosphate-glycosyltransferases (UGTs), and other enzymes [6,7,8,9].

Cytochrome P450s (CYPs) constitute a superfamily of hemoproteins and are pervasively distributed among animals, plants, fungi, and bacteria [10]. The cysteine residues in the structure of CYPs form a sulfhydryl group by linking to the iron atom in the heme, which together constitutes the structural center of CYPs. This region is defined as the heme-binding domain and serves as the distinguishing feature of CYP proteins [11]. The diversity of CYP enzymes includes a wide range of substrates that enable a multitude of distinct catalytic reactions [12,13]. The enzymes present in the extract of the antennae of *Phyllopertha diversa* were observed to rapidly degrade a bioalkaline pheromone. The enzyme was found to be sensitive to CYP inhibitors and requires coenzyme NADPH activation, which indicated that CYP had a degrading function [14]. The EAG assay demonstrated that following the suppression of the antenna-specific P450 gene *CYP4AW1* in *P. diversa*, the antenna’s response to pheromones was significantly reduced, indicating that the gene is involved in the recognition process of pheromones [15]. The recombinant protein of the gene *DponCYP345E2*, which is specifically expressed in the antennae of *Dendroctonus ponderosae*, has the capacity to metabolize volatiles from host plants [16]. The use of RNA interference technology and EAG technology demonstrated the role of *CYP4L4* in the recognition pheromone of *Spodoptera litura*, with high expression levels observed in the antennae [17]. The antennal-specific expression of *CYP6FD5* in *Locusta migratoria* was implicated in the recognition of volatiles from the host plants [18].

The red imported fire ant (RIFA, *Solenopsis invicta* Buren), has been identified by the International Union for Conservation of Nature (IUCN) as one of the 100 most destructive invasive species. It is renowned for its aggressive and territorial behavior, which results in significant losses to the ecological diversity and economic stability of invaded regions [19,20]. *S. invicta* relies on its highly coordinated social behavior to obtain the energy it requires for essential life activities. The foundation for sustaining this sophisticated social conduct is the red fire ant’s intricate and multifaceted pheromones. Pheromones are indispensable for the maintenance of the social structure of the ant colony and for the survival and reproduction of the species. The alarm pheromone component of the *S. invicta* has been identified as 2-ethyl-3,6(5)-dimethylpyrazine, EDMP. When the *S. invicta* is disturbed or threatened, the mandibular glands of the workers secrete alarm pheromones to signal to nearby colony members to seek refuge [21,22,23]. Nevertheless, the existing literature on the subject indicates that there is a paucity of research investigating the role of the CYP genes in the olfactory recognition of alarm pheromones.

The objective of this study is to elucidate the role of highly expressed CYP genes in olfactory recognition of alarm pheromones in *S. invicta* antennae, we initially identified six highly abundant CYPs genes from *S. invicta* antennae transcriptome data. We focused our study on the two highly expressed CYPs genes in the antennae by quantitative real-time PCR (RT-qPCR): *SinvCYP6K1* and *SinvCYP4V2*. We used further RNAi technology and EAG to study the function of these two CYPs gene. Moreover, this study pioneers the use of DeepLabCut *v*.2.3.10 software to record and analyze the movement patterns of the *S. invicta*. These results provide evidence that the CYPs genes of *S. invicta* antennae provide recognition of alarm pheromones and enrich the mechanism of the ant’s sensing pheromone. This provides a theoretical basis for the development of techniques for pest management.

## 2. Materials and Methods

### 2.1. Insect Rearing

Complete *S. invicta* nests were excavated from Fujian Agriculture and Forestry University (26°5′26″ N, 119°14′4″ E) in April 2023 and Qishan Lake Park (26°2′40″ N, 119°11′46″ E) in Minhou County, Fuzhou City, China in June 2023, and reared under indoor conditions (temperature: 25 °C, light: 12 L:12 D, humidity controlled with a water-soaked skimmed cotton balls to approximately 80%, and ants fed daily with fresh yellow mealworm larvae and artificial food).

### 2.2. Sequencing of the Transcriptome

The antennae, head, thorax, and abdomen of *S. invicta* worker were later collected in 1.5 mL enzyme-free tubes containing RNA later^TM^ Solution (AM7020, Invitrogen. Thermo Fisher Scientific, Waltham, MA, USA), set up with three biological replicates, and placed in the refrigerator at 4 °C for overnight resting, then in the refrigerator at −80 °C. After the samples were collected, they were sent to Shanghai Ouyi Biologicals Co. Ltd. (Shanghai, China). for second-generation high-throughput sequencing.

Worker samples were subjected to RNA extraction, followed by DNA digestion using DNase. mRNA was enriched with magnetic beads coated with Oligo(dT). Fragmentation reagent was added to break the mRNA into short segments. After the constructed library passed quality control with the Agilent 2100 Bioanalyzer (Agilent Technologies, Santa Clara, CA, USA), sequencing was carried out using the Illumina HiSeqTM 2500 sequencer (Illumina, Greenland, NH, USA) to generate 125 bp or 150 bp paired-end data. Once quality control was passed, sequencing was conducted again using the Illumina sequencer.

### 2.3. Antennae Transcriptome CYPs Identification and Molecular Characterization

A database composed of known reference genome and annotation files, the htseq-count [24], was utilized to acquire the number of reads aligned to coding genes in each sample. Gene expression levels are determined using the FPKM [25] method (Fragments Per Kilobase of transcript per Million mapped reads), which quantifies the number of fragments originating from a protein-coding gene per kilobase pair of gene length, relative to the total number of fragments sequenced in millions. Expression analysis of genes annotated as cytochrome P450 in the sequenced samples was performed using FPKM values (with three replicates per sample), and a heatmap was constructed using the OECloud tools (https://cloud.oebiotech.com (accessed on 2 August 2024)). Protein molecular weights and isoelectric points were calculated using ExPASy (The SIB Swiss Institute of Bioinformatics) (http://www.ExPASy.org (accessed on 2 August 2024)).

### 2.4. Multiple Sequence Comparison and Phylogenetic Analysis

Genes annotated as cytochrome P450 (CYP) were found through analysis of various databases in transcriptome data. Gene sequences were obtained from NCBI and then compared using Clustal W (https://www.genome.jp/tools-bin/clustalw (accessed on 15 August 2024)) for multiple sequence comparison. Finally, the phylogenetic tree was constructed using the maximum likelihood method in MEGA5.2 software, and the values on the nodes indicate the percentage based on 1000 replicates.

### 2.5. Extraction of Total RNA and Synthesis of cDNA

Total RNA was extracted from four tissues (antennae, head, thorax and abdomen) of *S. invicta* according to the instructions of the total RNA extraction kit (SteadyPure RNA Extraction Kit, AG21024, Accurate Biotechnology, Changsha, China). cDNA purity was determined using Nanodrop 2000 (NanoDrop, Wilmington, NC, USA). RNA concentration, OD260/OD280 and OD260/OD230 values were measured using Nanodrop 2000 (NanoDrop, Wilmington, NC, USA), and RNA purity was measured using 1% agarose gel. cDNA was synthesized from 1 μg/μL RNA using a 5 × Evo M-MLV RT Reaction Mix Ver. 2 (Evo M-MLV RT Mix Kit with Gdna Clean, AG11728, Accurate Biology).

### 2.6. The Tissue and Developmental Stages Expression of CYPs

To understand the tissue and developmental expression profiles of the candidate CYPs genes, 100 workers were selected to dissect their antennae, head, thorax, and abdomen and samples of different ages of workers (egg, 4th instar, pre-pupa, pupa, adult, four biological replicates per tissue and stage) were collected and placed in RNA later^TM^ Solution (AM7020, Invitrogen, Thermo Fisher Scientific). We extracted RNA from each sample and synthesized cDNA. The fluorescence quantitative reaction was performed using a 7500 real-time fluorescence quantitative PCR instrument (Singapore); 3 technical replicates were used for each biological replicate, and the reaction system was as follows: 2 × SYBR Green qRT-PCR (AG11702, Accurate Biotechnology), 10 μL; RNase H_2_O, 7 μL; forward primers 1 μL; reverse primers 1 μL; cDNA 1 μL. The reaction program was: 95 °C, 2 min; 95 °C, 5 s; 60 °C, 30 s; 40 cycles. *EF1-β* was used as an internal reference (GenBank accession number: EH413796) [26], and relative expression was calculated using the comparative Ct method (2^−∆∆CT^) [27]. The specific primers used for RT-qPCR are listed in Appendix A.

Gene sequences were obtained from the *S. invicta* genome database (https://www.ncbi.nlm.nih.gov/, accessed on 2 August 2024; *SinvCYP6K1*: XM_039456789.1; *Sinv4V2*: XM_011171459.3), specific primers (Appendix A) were designed in SnapGene, and simple PCR was performed using an Eppendorf AG PCR instrument (Eppendorf, Hamburg, Germany) with adult workers antennae cDNA as template; the PCR reaction system was: 2 × Pro Taq Master Mix (dye plus) (AG11109, Accurate Biotechnology), 25 μL; ddH_2_O, 18 μL; forward primers, 2.5 μL; reverse primers, 2.5 μL; and cDNA, 2 μL. The PCR reaction program was: 95 °C, 3 min, 1 cycle; 95 °C, 30 s, 35 cycles; 55 °C, 30 s, 35 cycles; 72 °C, 2 min, 35 cycles; 72 °C, 5 min, 1 cycle, 12 °C, ∞.

### 2.7. RNA Interference

dsRNA (containing the target gene and GFP) was synthesized using the T7 RiboMAX™ Express RNAi System (P1700, Promega, Madison, WI, USA) according to the instructions. The T7 promoter-specific primers containing the T7 promoter used to synthesize dsRNAs using the cloned and sequenced correct plasmids as templates are shown in Appendix A. The dsGFP (GenBank accession number: ACY56286) was used as a control for all RNAi experiments. The concentration of dsRNA was determined using a NanoDrop 2000 spectrophotometer and its purity was detected on a 1.0% agarose gel. The dsRNA was mixed with distilled water and fed to worker ants [28]; 80 μL of 3 μg/μL of dsRNA was fed to each 80 workers. All tests were run in triplicate, with 20 workers in each replicate. The expression levels of the target genes in workers were detected using RT-qPCR at 12 h, 24 h, and 36 h after interference.

### 2.8. Electroantennography and Behavioral Assays

After 24 h of interference, the response of the workers to 2-ethyl-3,6(5)-dimethylpyrazine (EDMP) was detected by EAG; EDMP was set 3 concentrations (0.1 μg/μL, 1.0 μg/μL, 10.0 μg/μL) [29,30]. The heads of the workers were quickly cut off with a scalpel and placed between two glass electrodes connected to a conductive gel (Spectra 360 Electrode Gel). A volume of 20 μL of the odor stimulus source was added to a strip of filter paper (0.5 × 1.0 cm) and placed in a Pasteur pipette. The humidified airflow was delivered by a Syntech stimulus controller (model CS-55, Syntech, Dortmund, Germany) at a speed of 50 cm/s. The airflow was then adjusted to the temperature of the filter paper. EAG responses of the worker ant antennae to the odor compounds were recorded. Each sample was tested with hexane as a control, and EAG responses were recorded at 60 s intervals for each biological replicate of the test; all tests were run in triplicate.

We used DeepLabCut to perform behavioral assessments [31]. The behavioral responses of single-headed worker ants to 0.1 μg/μL, 1.0 μg/μL, and 10.0 μg/μL EDMP were recorded individually in each video. Three replicates were set for each concentration. The training dataset underwent 100,000 iterations, resulting in a training error of 2.56 pixels and a testing error of 3.32 pixels, with a cut-off value of 0.6. After successfully training the network, we applied it to analyze all the videos in the dataset. Python *v*.2.6 scripts were then used to analyze the behavior of each animal based on the results of the DeepLabCut *v*.2.3.10 model.

### 2.9. Statistic Analysis

Data were presented as mean ± standard deviation (SD). Analysis of one-way variance (ANOVA) followed by Tukey’s HSD test (*p* < 0.05) was used to analyze the tissue and developmental stage expressions of *SinvCYP* gene, and the workers’ EAG responses and trajectory behaviors. The silencing efficiency of ds*SinvCYP* and ds*GFP* were conducted using Student’s *t*-test (* *p* < 0.05, ** *p* < 0.01, *** *p* < 0.001). The data underwent analysis using the SPSS software (*v*. 23.0; SPSS Inc. Armonk, NY, USA), and the results of the statistical analyses were subsequently graphed using GraphPad Prism 9.0.

## 3. Results

### 3.1. SinvCYPs from Transcriptome

A screening of the antennal transcriptome of *S. invicta* worker was conducted to identify 95 genes that were annotated as CYPs (Appendix A). The seven expression levels of highly expressed CYPs in worker antennae were quantified, with FPKM values ranging from 364.35 to 1606.97. These FPKM levels of antennae were found to be higher than those observed in other tissues (Appendix A). Theoretical molecular weights were observed to fall within the range of 58.305–123.571 KDa, while isoelectric points were noted to fall within the range of 6.39–9.57 (Table 1). LOC105197405 exhibited the highest FPKM value within the transcriptome. However, available studies indicate the potential for a deletion in this sequence region, which encompasses the genomic nucleotide sequence of the start and stop codons. Consequently, the current information about this sequence is incomplete [32].

### 3.2. Sequence Characterization and Phylogenetic Analysis of SinvCYPs

A multiple sequence comparison of the amino acid sequences of SinvCYP6K1, SinvCYP6K1-1, SinvCYP4C1, SinvCYP4C1-1, SinvCYP4C1-2, and SinvCYP4V2, as previously mentioned, revealed that they all contain five conserved regions. The aforementioned regions include Helix C (Helix C, WXXXR), Helix I (Helix I, GXE/DTT/S), Helix K (Helix, EXXR), Meander (PXXFXPEX/DF), and heme-binding domain (PFXXGXRXCXG/A) [11] (Figure 1). To further characterize the phylogenetic relationships of these CYPs, they were clustered with those of *S. litura*, *Tribolium castaneum*, *Drosophila melanogaster*, *Apis mellifera*, *Dastarcus helophoroides*, *D. ponderosae*, *L. migratoria*, and *Acyrthosiphon pisum*. The resulting clusters were then analyzed, and the results demonstrated that SinvCYP4C1, SinvCYP4C1-1, SinvCYP4C1-2, and SinvCYP4V2 were found to cluster within the CYP4 family, exhibiting high levels of homology. It is therefore hypothesized that they may possess similar functional characteristics. SinvCYP6K1 and SinvCYP6K1-1 were identified as belonging to the CYP3 family, with SinvCYP6K1 displaying greater homology to AmelCYP6K1 of *A. mellifera*, which was a member of the *Hymenoptera* order. Additionally, CYP6K1-1 exhibits greater homology to Dpon345E2 of *D. ponderosae*, which is a gene that is specifically expressed in the antennae of *D. ponderosae* (Figure 2).

### 3.3. The Tissue and Developmental Stages Expression of SinvCYPs

Tissue expression profiles demonstrated that SinvCYP6K1 and SinvCYP4V2 were markedly elevated in the antennae relative to other tissues. Their expression in the head, thorax, and abdomen was notably low (*p* < 0.001) (Figure 3A,F). All six CYPs exhibited the highest expression in the antennae, SinvCYP6K1-1, SinvCYP4C1, SinvCYP4C1-1, and SinvCYP4C1-2 were all observed to be expressed in the head, thorax, and abdomen of worker, and were not found to be specifically expressed in the antennae (Figure 3B–E).

A further developmental expression analysis of *SinvCYP6K1* and *SinvCYP4V2*, which are highly expressed in the antennae of worker, revealed that both genes exhibited the highest expression levels in the adult stage of red fire ants, with almost no expression observed in the egg, larval, and nymphal stages (*p* < 0.001) (Figure 4A,B). Thus, it is postulated that these two genes play a pivotal role in the olfactory function of worker and may be implicated in the process of pheromone recognition in *S. invicta*.

### 3.4. RNAi Effect of SinvCYPs

The results of the experiment demonstrated a notable decline in the expression of *SinvCYP6K1* of *S. invicta* following the administration of ds*CYP6K1* for both 12 and 24 h. In comparison to the control group that was fed ds*GFP*, the expression of *SinvCYP6K1* in *S. invicta* exhibited a significant reduction of 71.30% and 60.23% at 12 and 24 h respectively (12 h: *p* < 0.001; 24 h: *p* = 0.004 (Figure 5A); *S. invicta* fed ds*CYP4V2* for 24 h exhibited a significant decrease in *SinvCYP4V2* expression of 66.67% (*p* = 0.018) (Figure 5B); ds*CYP6K1* and ds*CYP4V2* mixed feeding for 12 h and 24 h resulted in a significant decrease in *SinvCYP6K1* and *SinvCYP4V2* expression. The expression of *SinvCYP6K1* was found to decrease significantly by 90.22% and 65.55% (12 h: *p* = 0.002; 24 h: *p* = 0.015) (Figure 5C), while ds*CYP6K1* and ds*CYP4V2* mixed feeding for 24 h resulted in the expression of *SinvCYP4V2* showing a significant decrease of 87.87% (*p* = 0.007) compared to ds*GFP* (Figure 5D).

### 3.5. The EAG Response and Behavioral Assays of RIFA to EDMP by RNAi

Worker ants fed with ds*CYP6K1*, ds*CYP4V2*, and ds*CYP6K1 + CYP4V2* resulted in a markedly diminished EAG response to EMDP in the workers when compared to the feeding of ds*GFP* (0.1 μg/μL: *p* < 0.001). The EAG responses of the treatment groups to EMDP were significantly different (*p* < 0.001) at concentrations of 0.1, 1.0, and 10.0 μg/μL, while the EAG responses to hexane did not change significantly (Figure 6). Following the disruption of *SinvCYP6K1* and *SinvCYP4V2*, behavioral trajectory imaging revealed a reduction in the range of movement and distance covered by the workers (Figure 7B), accompanied by a notable decline in their movement rate compared to that observed in ds*GFP*-fed workers (Figure 7A).

## 4. Discussion

In the present study, RNA sequencing technology is used to identify six cytochrome P450 (CYP) genes with high expression levels in the antennae of *S. invicta* worker ants. The RT-qPCR analysis validated the highly expression of *SinvCYP6K1* and *SinvCYP4V2* in the antennae of *S. invicta*. To elucidate the functional role of these genes in red fire ant recognition of alarm pheromones, we employed RNA interference (RNAi) technology in conjunction with EAG analysis and DeepLabCut *v*.2.3.10 software. The results of this study demonstrated that *SinvCYP6K1* and *SinvCYP4V2* were involved in the recognition process of alarm pheromone.

The substantial literature base indicates that the cytochrome P450 family of genes plays a significant role in insect chemical communication, particularly with regard to CYPs, which are highly expressed in antennae and may serve a pivotal function in olfactory signaling and processing. Evidence from a number of studies indicates that CYPs are responsible for the inactivation of odorous molecules and pheromones in a diverse array of insects [14,15]. The structure of the CYP gene was composed of five conserved regions [11]. In this study, the six highly expressed SinvCYPs and the *S. litura*, *A. mellifera*, and *D. ponderosae* CYPs were found to contain similar specific conserved regions. However, the amino acids within these conserved regions differed, suggesting that their specific functions may have been altered [33]. In this study, the six highly expressed SinvCYPs and the CYPs from *S. litura*, *A. mellifera*, and *D. ponderosae* all contained similar specific conserved regions, but the amino acids within the conserved regions differed, suggesting that their specific functions may had changed [11,34]. Phylogenetic analysis of *SinvCYPs* with six other insect species showed that *SinvCYP4C1*, *SinvCYP4C1-1*, *SinvCYP4C1-2*, and *SinvCYP4V2* clustered in the CYP4 family, whereas *SinvCYP6K1* and *SinvCYP6K1-1* clustered in the CYP3 family. Research was shown that the CYP3 family is mainly involved in the metabolism and detoxification of exogenous compounds in insects, and these enzymes may help insects cope with toxic chemicals in the environment and protect them from harmful compounds [35], while the CYP4 family may be involved in a wider range of biosynthetic and metabolic processes in insects, including olfactory related compounds, which may play a role in regulating olfactory receptors, recognizing odor molecules, and signaling [36]. The olfactory functions of the CYP3 and CYP4 families have been reported in insects; e.g., the mountain pine beetle *DponCYP345E2* reacts with host plant volatiles, enabling the borer to resist plant defense compounds [16,37], and the inhibition of the antennal *CYPAW1* enzyme in *P. diversa* by P450 inhibitors had been observed to result in a reduction in its olfactory sensitivity to sex pheromones [15]. In light of these findings, we postulate that the exclusively high expression of CYPs in the antennae of *S. invicta* may play a pivotal role in olfactory processes.

The expression pattern of CYPs may reflect their underlying biological functions, and their expression in different tissues corresponds to different biological functions [38,39,40]. The abdomen and fat body were the primary organs responsible for detoxification in most species. Therefore, the predominant CYPs in these organs were associated with detoxification metabolism in insects, as evidenced in the scientific literature [41,42,43]. In contrast, antennal-specific CYPs are linked to olfactory functions [32]. The tissue expression analysis of six antennal-specific CYPs revealed that *SinvCYP6K1* and *SinvCYP4V2* were specifically expressed in the antennae. Furthermore, the developmental expression analysis of *SinvCYP6K1* and *SinvCYP4V2* demonstrated that they were only expressed in adult worker, indicating that they may possess distinctive olfactory functions. The antennae of *Mamestra brassicae* have been observed to express *CYP4S4* and *CYP4L4* at a significantly higher level than other regions of the insect. In situ hybridization results indicated that the two P450 genes in question were localized to the antennal trichodea sensilla, which are responsible for odor molecule recognition. It had been hypothesized that these two genes may function as degradation of odorant substances in the antennae [44]; the antennal-specific expression of *CYP4C99*, *CYP6FD5*, *CYP6NY1*, and *CYP3327A1* of *L. migratoria*, was investigated to ascertain their potential involvement in the olfactory process of recognizing volatiles from host plants by *L. migratoria* [45]. In *S. invicta* colonies, adult worker ants are mainly responsible for nest defense and foraging. Therefore, the highly expressed CYPs genes in the antennae of adult workers are more important. It is speculated that *SinvCYP6K1* and *SinvCYP4V2*, which are highly expressed in the antenna, may play an important role in the olfactory function of *S. invicta*.

The alarm pheromone of *S. invicta* has the effect of maintaining a state of high alertness among members of the ant colony. Concurrently, individuals that receive the alarm signal will promptly proceed towards the source of the signal or flee collectively [46,47,48]. The repression of CYP expression, which is associated with olfactory function, may result in a reduction in the sensitivity of olfactory neurons, consequently diminishing the behavioral response of *S. invicta* to pheromone. To gain further insight into the role of *SinvCYP6K1* and *SinvCYP4V2* in olfactory recognition of EDMP, we employed the method of feeding dsRNA to *S. invicta* through an aqueous solution. This approach led to a notable reduction in the expression of *SinvCYP6K1* and *SinvCYP4V2* in the antenna of worker ants, suggesting that this method could be utilized for the treatment of large populations of *S. invicta*. Single or double interference of *SinvCYP6K1* or *SinvCYP4V2* resulted in a notable reduction in the EAG response of *S. invicta* to EDMP. Additionally, the range and speed of movement in response to EDMP were also significantly diminished. The present findings indicate that *SinvCYP6K1* and *SinvCYP4V2* are implicated in the recognition of EDMP by *S. invicta*. Similar studies have also been conducted. For example, in the case of *S. litura*, the interference with the highly expressed antennal *CYP4L4* resulted in a significant decrease in the EAG response to sex pheromones [17]. In *L. migratoria*, the interference with the antennal-specific expression of *LmCYP6FD5* resulted in a notable reduction in the EAG response to the volatiles of the grass family host plant, suggesting that *LmCYP6FD5* may play a role in the recognition of host volatiles [18]. There are many types of ODEs in the insect olfactory system, and different ODEs can work together in insects to recognize the same odor molecule [49]. In previous studies, a clear dose–effect relationship was observed between the EAG response of the antennal sensilla of *S. invicta* to EDMP. Specifically, the higher the concentration of EDMP, the stronger the EAG response [50]. However, in the present study, *S. invicta* exhibited no discernible dose–effect response to varying concentrations of EDMP following disturbance. This may be attributed to the diminished olfactory capacity of the worker ants following disturbance, which demonstrated a consistently low EAG response to EDMP across all concentrations.

The present study identified the cytochrome P450s *SinvCYP6K1* and *SinvCYP4V2*. It is currently unclear whether these enzymes are capable of reacting with EDMP and, if so, the mechanism by which such a reaction would proceed. Therefore, further study is necessary to investigate the interaction between *SinvCYP6K1* and *SinvCYP4V2* protein and EDMP. The recombinant proteins can be purified via in vitro expression, and their enzymatic activity can be quantified. Additional research utilizing high-performance liquid chromatography (HPLC) technology can be conducted to ascertain whether the recombinant proteins of SinvCYP6K1 and SinvCYP4V2 are capable of degrading EDMP in vitro [51], thus establishing a novel theoretical foundation for a more profound comprehension of the olfactory recognition and degradation of pheromones by *S. invicta*.

## Figures and Tables

**Figure 1 insects-16-00043-f001:**
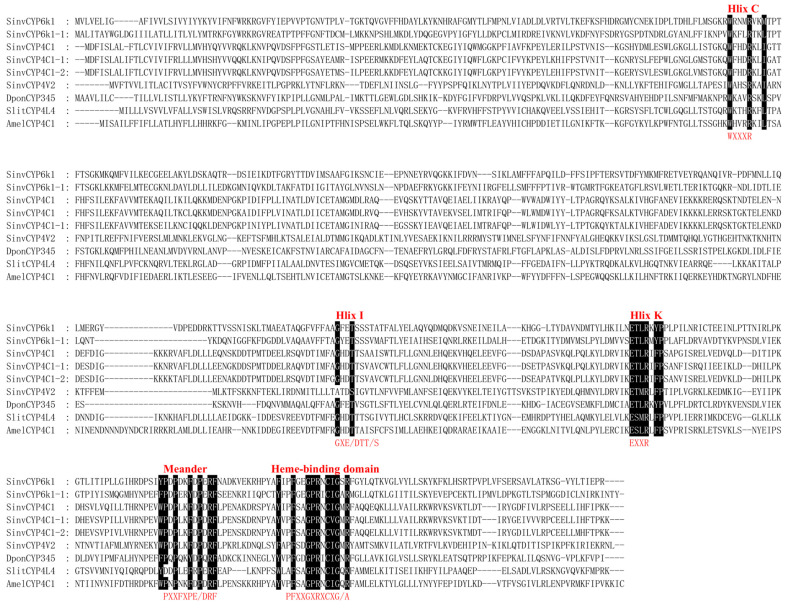
Amino acid sequence analysis of CYPs from SinvCYP4C1, SinvCYP4C1-1, SinvCYP4C1-2, SinvCYP4V2, SinvCYP6K1, and SinvCYP6K1-1, and their comparison with CYPs from *S. litura*, *A. mellifera*, and *D. ponderosae*. Black-labeled regions represent cytochrome P450 conserved regions, which include the following patterns: Helix C: WXXXR, Helix I: GXE/DTT/S, Helix: EXXR, Meander: PXXFXPEX/DF, and heme-binding domain: PFXXGXRXCXG/A. Gene bank accession numbers of other insect CYPs utilized for amino acid sequence analysis are presented in Appendix A.

**Figure 2 insects-16-00043-f002:**
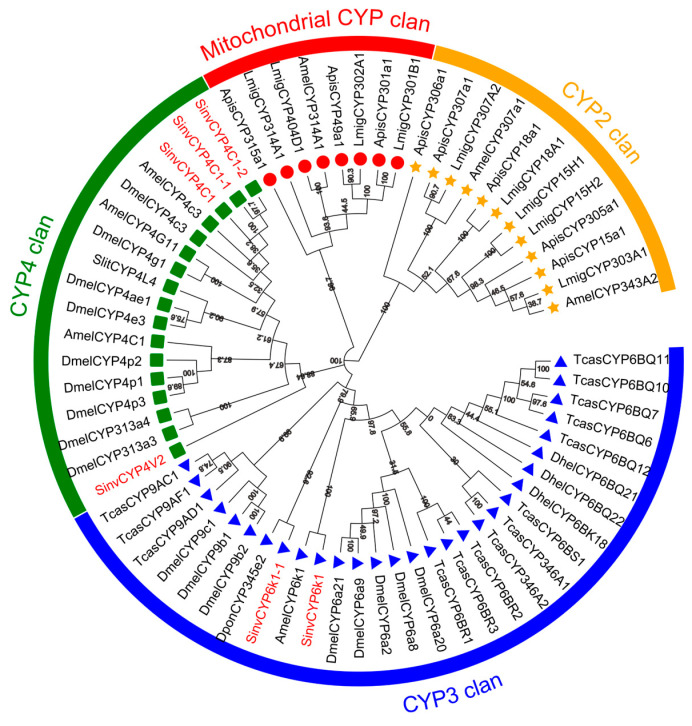
Phylogenetic analysis of SinvCYP4C1, SinvCYP4C1-1, SinvCYP4C1-2, SinvCYP4V2, SinvCYP6K1, and SinvCYP6K1-1 in conjunction with CYPs from other insects (blue indicates CYP3 family, green indicates CYP4 family, red indicates mitochondrial CYP family, and yellow indicates CYP2 family). The resulting developmental trees were constructed using the maximum likelihood method, with values indicating percentages based on 1000 replicates. Gene bank accession numbers of other insect CYPs utilized for phylogenetic analysis are presented in Appendix A.

**Figure 3 insects-16-00043-f003:**
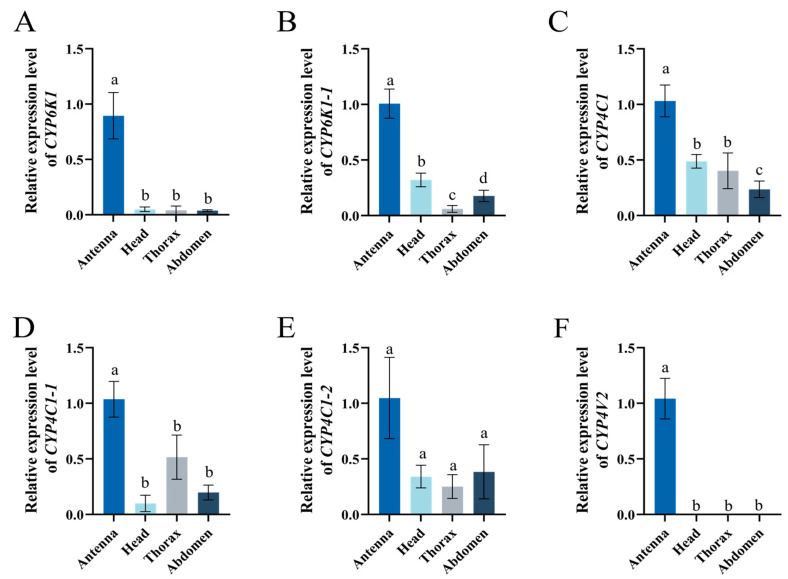
Expression pattern of *SinvCYPs* in antenna, head, thorax, and abdomen of *S. invicta*. (**A**) Expression of *SinvCYP6K1*. (**B**) Expression of *SinvCYP6K1-1*. (**C**) Expression of *SinvCYP4C1*. (**D**) Expression of *SinvCYP4C1-1*. (**E**) Expression of *SinvCYP4C1-2*. (**F**) Expression of *SinvCYP4V2*. Columns and error bars represent mean ± SD (n = 4). Different letters above bars indicate significant differences between different tissues according to Tukey’s HSD test (one-way ANOVA, *p* < 0.05).

**Figure 4 insects-16-00043-f004:**
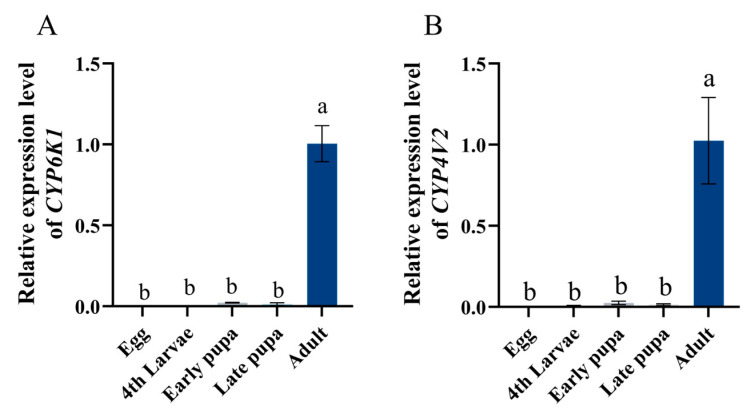
Expression patterns of *SinvCYP6K1* and *SinvCYP4V2* at different developmental stages. (**A**) Expression of *SinvCYP6K1*. (**B**) Expression of *SinvCYP4V2*. Mean ± SD (n = 4). Different letters above bars indicate significant differences between different developmental stage according to Tukey’s HSD test (one-way ANOVA, *p* < 0.05).

**Figure 5 insects-16-00043-f005:**
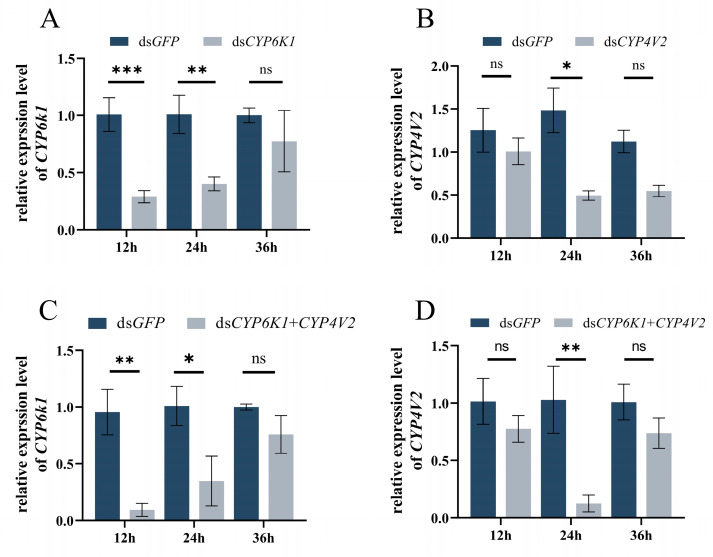
Interference efficiency of *SinvCYP6K1* and *SinvCYP4V2* following 12, 24, and 36 h of feeding specific dsRNAs. (**A**) Expression of *SinvCYP6K1* following feeding of ds*CYP6K1*. (**B**) Expression of *SinvCYP4V2* following feeding of ds*CYP4V2*. (**C**) Expression of *SinvCYP6K1* after mixed feeding of ds*CYP6K1* and ds*CYP4V2*. (**D**) Expression of *SinvCYP4V2* after mixed feeding of ds*CYP6K1* and ds*CYP4V2*. Data are means ± SD of 3 biological repeats. Asterisks (*) on bars indicate significant differences (ns, *p* > 0.05, * *p* < 0.05, ** *p* < 0.01, *** *p* < 0.001, Student’s *t*-test).

**Figure 6 insects-16-00043-f006:**
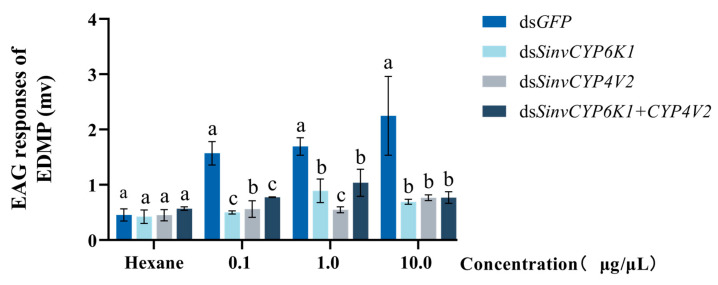
Response of workers to varying concentrations of EDMP following 24 h of dsRNA feeding. Concentrations tested were 0.1, 1.0, and 10.0 μg/μL. Columns and error bars represent mean ± SD (n = 3). Different letters above bars indicate significant differences between different tissues according to Tukey’s HSD test (one-way ANOVA, *p* < 0.05).

**Figure 7 insects-16-00043-f007:**
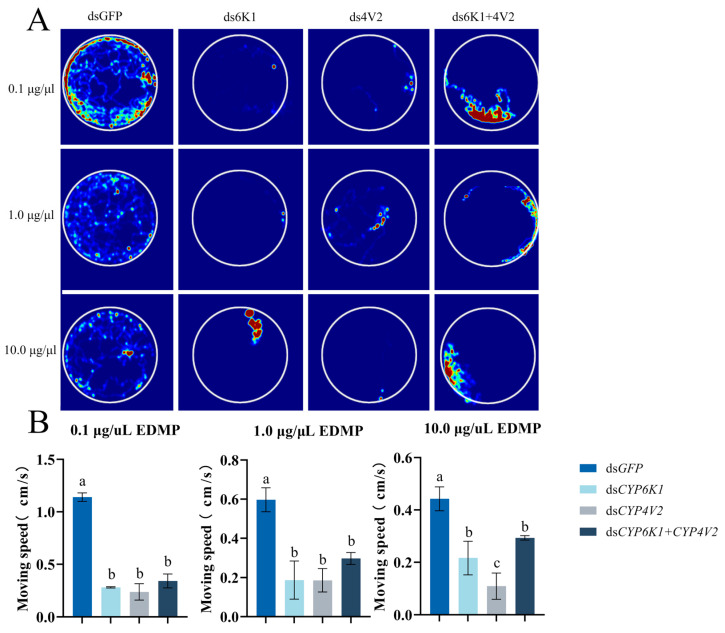
Behavioral response of workers to exposure to EDMP was assessed following a 24 h feeding period with double-stranded RNA (dsRNA). (**A**) Trajectory thermograms of *S. invicta* fed ds*GFP*, ds*CYP6K1*, ds*CYP4V2*, and ds*CYP6K1 + 4V2* for 24 h in response to 0.1, 1.0, and 10.0 μg/μL EDMP. (**B**) Movement velocities of worker ants in response to 0.1, 1.0, and 10.0 μg/μL EDMP after 24 h of feeding with ds*GFP*, ds*CYP6k1*, ds*CYP4V2*, and ds*CYP6K1 + 4V2*. Columns and error bars represent mean ± SD (n = 3). Different letters above bars indicate significant differences between different tissues according to Tukey’s HSD test (one-way ANOVA, *p* < 0.05).

**Table 1 insects-16-00043-t001:** The properties of the CYPs that are highly expressed in the antennae of *S. invicta*.

GenBank Accession Number	Gene Name	Amino Acid	MW (KDa)	PI
XM_039456789.1	*CYP6K1*	506	58.398	8.99
XM_039459678.1	*CYP6K1-1*	512	58.941	6.92
XM_026131197.2	*CYP6K1-2*	1064	123.571	8.64
XM_011171510.3	*CYP4C1*	510	58.929	6.63
XM_026134986.2	*CYP4C1-1*	512	59.333	7.97
XM_026134988.2	*CYP4C1-2*	512	59.305	6.39
XM_011171459.3	*CYP4V2*	501	58.974	9.57

Note: MW represents molecular weight; PI represents isoelectric point.

## Data Availability

Data are contained within the article and Appendix A.

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
