# Peer review of "High Antennal Expression of CYP6K1 and CYP4V2 Participate in the Recognition of Alarm Pheromones by Solenopsis invicta Buren"

_insects, 2025, doi:10.3390/insects16010043_

Round 1
Reviewer 1 Report
Comments and Suggestions for Authors
Line 40 to 45 – There is duplicated information, so I suggest rewrite these paragraph. Moreover, ODEs is the acronym for odorant-degrading enzymes. Several enzymes can be grouped in this term, for instance, esterases (CXEs), aldehyde oxidases (AOXs), alcohol dehydrogenases (ADs), cytochrome p450s (CYPs).
Line 89 – Hypothesis should be written right before the objective.
In methodology (2.4), you should use at least two housekeeping gene as an internal reference.
Lines 161-162 – Be careful with the temperatures and timing.
I suggest to refer Solenpsis invicta as S.invicta in the whole text instead RIFA.
In methodology (2.6), which was the concentration of the pheromone used?
You have to choose whether to use qRT-PCR or RT-qPCR.
Sometimes you say that SinvCYP6K1 and SinvCYP4V2 had a specific expression in antennae, however qRT-PCR results (Fig. 3) showed that are expressed in the other tissues. Therefore, I suggest do not use the term “specific”, it could be “highly” or “mainly” expressed in the antennae. This also has to be related to the hypothesis.
Line 340 – wrong format citation
If a specie is mentioned for the first time you have to write the full name, then you have to abbreviate it, for instance, Spodoptera litura and S. litura.
Line 370 and 416 – In situ and in vitro in italics.
Author Response
Response to Reviewer 1 Comments
- Summary
Thank you very much for the valuable feedback provided on our study. Over the course of the past ten days, we have extensively and carefully revised the manuscript, taking your suggestions into account. We have marked the revisions in the manuscript with corresponding colors, as indicated in the attached file. In our detailed responses, provided separately, we have addressed each reviewer's comments individually. We greatly appreciate your consideration of our revised manuscript and look forward to receiving the final decision from the editor and reviewers.
Response: We would like to thank the reviewer for their valuable feedback on our manuscript. We were grateful for the positive remarks and for pointing out areas where further explanation was needed to enhance the clarity of our experiments. We appreciated the opportunity to address these concerns before accepting the paper. We had carefully reviewed the comments and had made the necessary revisions accordingly.
- Point-by-point response to Comments and Suggestions for Authors
Comments 1: Line 40 to 45 - There is duplicated information, so I suggest rewrite these paragraph. Moreover, ODEs is the acronym for odorant-degrading enzymes. Several enzymes can be grouped in this term, for instance, esterases (CXEs), aldehyde oxidases (AOXs), alcohol dehydrogenases (ADs), cytochrome p450s (CYPs).
Response 1: Thank you for your suggestion, We have reorganized the section and the redundancy has been removed, you can see my changes in line 44-45. Also, the enzymes esterases (CXEs), aldehyde oxidases (AOXs), alcohol dehydrogenases (ADs), cytochrome p450s (CYPs) are part of the ODE, and we mentioned that the ODE includes these enzymes in the paragraph in line 55-57, to lead into our next paragraph.
Comments 2: Line 89 - Hypothesis should be written right before the objective.
Response 2: We are grateful for your insightful feedback. In light of your recommendations, we have revised this section and updated the original text accordingly. The revised text is as follows: “The objective of this study is to elucidate the role of highly expressed CYP genes in olfactory recognition of alarm pheromones in S. invicta antennae, we initially identified 6 highly abundant CYPs gene from S.invicta antennae transcriptome data. We focused our study on the two highly expression CYPs gene in the antennae by quantitative real-time PCR (RT-qPCR): SinvCYP6K1, SinvCYP4V2. We used further RNAi technology and EAG to study the function of these two CYPs gene. Moreover, this study pioneers the use of DeepLabCut software to recorded and analyzed the movement patterns of the S.invicta. These results provide evidence for S.invicta antennae CYPs recognition of alarm pheromones and enriched the mechanism of ant’s sensing pheromone. it provides a theoretical basis for the development of techniques for the pest management. Meantime, it provides a theoretical basis for the development of techniques for the pest management.” The modifications made by the author can be observed in line 90-98.
Comments 3: In methodology (2.4), you should use at least two housekeeping gene as an internal reference.
Response 3: In this study, ef1-β was employed as an internal reference gene, and the resulting quantitative data more accurately reflected the expression trend of the target gene in the transcriptome data. Therefore, the experimental results were deemed credible. Furthermore, ef1-β had been demonstrated to be the most suitable reference gene for use in quantitative experiments of various genes at different developmental stages of S. invicta due to its stability (Cheng et al., 2013). In this study, we quantified the CYP genes in different developmental stages of S. invicta. Given that gene expression in S. invicta may be significantly influenced by developmental stages and different tissues (Tian et al., 2004; Ometto et al., 2011), the increased use of other unvalidated internal reference genes would not ensure their stability in different insect states, which could have a detrimental impact on the experimental results.
References:
Cheng, D.; Zhang, Z.; He, X.; Liang, G. Validation of reference genes in Solenopsis invicta in different developmental stages, castes and tissues. PLoS One 2013, 8, e57718.
Tian, H.S.; Vinson, S.B.; Coates, C.J. Differential gene expression between alate and dealate queens in the red imported fire ant, Solenopsis invicta Buren (Hymenoptera: Formicidae). Insect Biochem. Mol. Biol. 2004, 34, 937-49.
Ometto, L.; Shoemaker, D.; Ross, K.G.; Keller, L. Evolution of gene expression in fire ants: the effects of developmental stage, caste, and species. Mol. Biol. Evol. 2011, 28, 1381-92.
Comments 4: Lines 161-162 - Be careful with the temperatures and timing.
Response 4: We were grateful for your input. We had revised this section to provide greater clarity regarding the temperature and time parameters outlined in the methodology. Please refer to line 174-175 for details of the amendments we have made.
Comments 5: I suggest to refer Solenpsis invicta as S.invicta in the whole text instead RIFA.
Response 5: We were grateful for your valuable input regarding the specifics. In accordance with your recommendation, we had substituted RIFA in the complete text with the abridged designation of Solenopsis invicta: “S. invicta”.
Comments 6: In methodology (2.6), which was the concentration of the pheromone used?
Response 6: Thank you for posing this question. In the methodology 2.6, the warning pheromone EDMP exhibits three distinct concentration gradients, namely 0.1, 1.0 and 10.0 μg/μL. The selection of concentrations for candidate compounds (0.1, 1.0, 10.0 μg/μL) in the manuscript were primarily referred to the methodology of a behavioral experiment described in section 2.9 of a paper published in Pest Management Science: “Xu, C.; Yang, F.; Duan, S.; Li, D.; Li, L.; Wang, M.; Zhou, A. Discovery of behaviorally active semiochemicals in Aenasius bambawalei using a reverse chemical ecology approach. Pest Manag. Sci. 2021, 77, 2843-2853.”
Comments 7: You have to choose whether to use qRT-PCR or RT-qPCR.
Response 7: We were grateful for your suggestion. In accordance with your recommendation, the methodology presented in this article was “RT-qPCR”, and the full text had been meticulously examined to guarantee methodological consistency throughout.
Comments 8: Sometimes you say that SinvCYP6K1 and SinvCYP4V2 had a specific expression in antennae, however qRT-PCR results (Fig. 3) showed that are expressed in the other tissues. Therefore, I suggest do not use the term “specific”, it could be “highly” or “mainly” expressed in the antennae. This also has to be related to the hypothesis.
Response 8: Thank you for your suggestion. According to your advice, we have changed "specific" to "highly" in line 275.
Comments 9: Line 340 - wrong format citation
Response 9: We would like to express my gratitude for your meticulous attention to the text. The erroneous formatting of the citation had been rectified.
Comments 10: If a specie is mentioned for the first time you have to write the full name, then you have to abbreviate it, for instance, Spodoptera litura and S. litura.
Response 10: We were grateful for your expert counsel. In accordance with your guidance, we examined the initial appearance of the species in the full text and subsequently abbreviated it upon its subsequent occurrence.
Comments 11: Line 370 and 416 - In situ and in vitro in italics.
Response 11: We were grateful for your suggestions. We have corrected "In situ” and “in vitro" to italics in line 390.
Reviewer 2 Report
Comments and Suggestions for Authors
In the manuscript 'Antennal specific CYP6K1 and CYP4V2 participate in the recognition of alarm pheromones by Solenopsis invicta Buren', Jiang and colleagues identified six CYP450s overexpressed in the transcriptome of worker ant antennae. Their expression was confirmed by RT-qPCR and only two genes, CYP6K1 and CYP4V2, showed up-regulation in worker ant antennae and adults. These two genes were disrupted by RNAi and down-regulation was observed. Worker ants fed with dsCYP6K1 and dsCYP4V2 showed reduced antennal response and trajectory behavior to the alarm pheromone (2-ethyl-3,6(5)-dimethylpyrazine). Although the results of this article are relevant, a major revision is needed before publication. My main criticism is that the manuscript lacks an aim and/or hypothesis. The materials and methods are unclear, and it is difficult to follow the experimental procedure they used. The results allowed me to deduce the experimental procedure of the research. I suggest expanding the discussion section according to the results obtained. The manuscript needs a revision of the English language.
Major comments
Review nomenclature classification of the CYPs (Feyereisen, 2012).
Introduction. Include the hypothesis and/or objectives of the research.
Materials and Methods rewrite this section, based on the research objectives. I suggest the authors include separated sections for transcriptome antennae and Expression of CYP’s and detail in each one the procedure including references or links to each platform used.
For transcriptome indicated tissue or organ, number of replicates, sequencing platform, and transcriptome assembly and annotation parameters.
For expression by RT-qPCR indicated tissue or organ, stage of development, number of biological replicates, number of technical replicates, indicate the endogenous gene used and the method of analysis.
Minor comments
I suggest use up-regulation and down-regulation in exchange for high or low expression.
Line 12 and 15. Include worker ant.
Line 21 Change subfamilies by family.
Abstract. Include the objective of the research.
Line 28. Workers with or without dsCYP?
Line 44-45. This sentence has already been mentioned before, delete.
Line 53. Change ODE has by ODEs have…
Line 54. Change It is by They are…
Line 56. Change It includes by ODEs include..
Line 59. Change heme superfamily by superfamily of hemo proteins
Line 61. Include heme prosthetic group
Line 63. Include recognition of a wide range…
Line 64-69 rewrite the paragraph
Line 127. Include the names of the databases you used and the URL to them.
Line 128. What do you mean by sequence numbers of the genes.
Line 129. BLAST not defined when first mentioned
Line 131. Why neighbour-joining? I suggest use maximum likelihood method, include the access number of the sequences and species names used for phylogenetic reconstruction.
Line 131. Missing URL of MEGA5.2
Line 133. Separate the physico-chemical characteristics in another section.
Lines 134-138. Include this information in the expression of CYP’s section and add how to calculate of FPKM and differential expression?
Lines 140-144. What was the purpose of the cDNA synthesized from the section 2.1?
Line 154-155. Eliminate this sentence.
Line 155-162. What was the purpose of doing an end-point PCR?
Line 156. Insert the accession number of the genome.
Line 164. Please define GFP.
Line 172. Triplicate
Line 176. Which components?
Line 179. 20 mL or 20 uL?
Line 186-190. Please include what type of insects you use, ds-CYP fed or unfed. How many insects used?
Line 191. Improve the description of statistical analyses.
Line 202. Previously when??
Lines 202-204. This is an interpretation of results, review this section and just describe the results.
Table 1. How was assignment the CYP names? And It is correct the length of CYP6K1-2?
Line 218. Change Hlix K by K-Helix.
Line 222-224. This is information
Fig. 3. Please indicate the units of expression
Lines 277-279. Remove this paragraph.
Line 282, Fig. 5 and 6. Please use h by hours.
Line 294. Different font size.
Line 301. Workers ants fed with dsCYP6K1…..
Line 304. Why these concentrations? Include this information in Material and Methods
Line 311. Eliminate letter T.
Discussion section. Rebuild this section.
Line 340. This cite has a different format.
Comments on the Quality of English Language
The manuscript needs a revision of the English language.
Author Response
Response to Reviewer 2 Comments
- Summary
In the manuscript 'Antennal specific CYP6K1 and CYP4V2 participate in the recognition of alarm pheromones by Solenopsis invicta Buren', Jiang and colleagues identified six CYP450s overexpressed in the transcriptome of worker ant antennae. Their expression was confirmed by RT-qPCR and only two genes, CYP6K1 and CYP4V2, showed up-regulation in worker ant antennae and adults. These two genes were disrupted by RNAi and down-regulation was observed. Worker ants fed with dsCYP6K1 and dsCYP4V2 showed reduced antennal response and trajectory behavior to the alarm pheromone (2-ethyl-3,6(5)-dimethylpyrazine). Although the results of this article are relevant, a major revision is needed before publication. My main criticism is that the manuscript lacks an aim and/or hypothesis. The materials and methods are unclear, and it is difficult to follow the experimental procedure they used. The results allowed me to deduce the experimental procedure of the research. I suggest expanding the discussion section according to the results obtained. The manuscript needs a revision of the English language.
Response: Thank the reviewer for your valuable feedback on our manuscript. We sincerely appreciate these positive remarks and for pointing out areas where further explanation is needed to enhance the clarity of our experiments. We are grateful for the opportunity to address these concerns before accepting the paper. We have carefully reviewed the comments and have made the necessary revisions accordingly.
- Point-by-point response to Comments and Suggestions for Authors
Major Comments:
- Review nomenclature classification of the CYPs (Feyereisen, 2012).
Response: Thanks for your suggestion, we checked the naming classification of CYPs,we changed all ”CYP4 subfamily, CYP6 subfamily” to “CYP4 family, CYP6 family” in manuscript.
- Include the hypothesis and/or objectives of the research.
Response: We were grateful for your insightful feedback on the article. In light of your recommendations, we had revised the manuscript and included our research objective on line 90-91.
- Materials and Methods rewrite this section, based on the research objectives. I suggest the authors include separated sections for transcriptome antennae and Expression of CYP’s and detail in each one the procedure including references or links to each platform used.
Response: We were most grateful for your valuable comments. Based on your suggestions, we had rewritten Methods 2.2 and 2.3 to make each step more clearly delineated. The specific corrections can be seen in the original article, lines 108-131. Specific corrections as follows:
2.2 Sequencing of the transcriptom
The antennae, head, thorax and abdomen of S. invicta worker were later collected in 1.5 ml enzyme-free tubes containing RNA, set up with three biological replicates, and placed in the refrigerator at 4℃ for overnight resting, and then in the refrigerator at -80℃.After the samples were collected, they were sent to Shanghai Ouyi Biologicals Co. Ltd. for second-generation high-throughput sequencing.
Worker samples were subjected to RNA extraction, followed by DNA digestion using DNase. mRNA was enriched with magnetic beads coated with Oligo(dT). Fragmentation reagent was added to break the mRNA into short segments. After the constructed library passed quality control with the Agilent 2100 Bioanalyzer (Agilent Technologies, Santa Clara, CA, USA), sequencing was carried out using the Illumina HiSeqTM 2500 sequencer (Illumina, Greenland, NH, USA)to generate 125bp or 150bp paired-end data. Once quality control was passed, sequencing was conducted again using the Illumina sequencer.
2.3 Antennae transcriptome CYPs identification and molecular characterisation
Using a database composed of known reference genome and annotation files, the htseq-count (Anders et al., 2015) was utilized to acquire the number of reads aligned to coding genes in each sample. Gene expression levels are determined using the FPKM (Roberts et al., 2011) method (Fragments Per Kilobase of transcript per Million mapped reads), which quantifies the number of fragments originating from a protein-coding gene per kilobase pair of gene length, relative to the total number of fragments sequenced in millions. Expression analysis of genes annotated as cytochrome P450 in the sequenced samples was performed using FPKM values (with three replicates per sample), and a heatmap was constructed using the OECloud tools (https://cloud.oebiotech.com). Protein molecular weights and isoelectric points were calculated using ExPASy (The SIB Swiss Institute of Bioinformatics) (http://www.ExPASy.org).
- For transcriptome indicated tissue or organ, number of replicates, sequencing platform, and transcriptome assembly and annotation parameters.
Response: We were grateful for your valuable input. We had incorporated pertinent descriptions regarding the organization, number of experimental replicates, sequencing platforms, transcriptome assembly, and annotation parameters in Method 2.2.
- For expression by RT-qPCR indicated tissue or organ, stage of development, number of biological replicates, number of technical replicates, indicate the endogenous gene used and the method of analysis.
Response: Line 152-165 We described the RT-qPCR analysis of the expression levels of candidate CYPs in S. invicta worker tissues (antennae, head, thorax, and abdomen) and different developmental stages (egg, 4th instar, pre-pupa, pupa and adult), four biological replicates were set up for each tissue or developmental stage, with each biological replicate comprising three technical replicates. The ef1-β was used as the housekeeping gene in experiments (GenBank accession number: EH413796), and the relative expression was calculated using the 2-∆∆CT method.
Minor Comments:
- I suggest use up-regulation and down-regulation in exchange for high or low expression.
Response: Thank you very much for your comments, after our discussion, we thought that some up-regulated genes in the tentacles, but their expression was very low, may not play the key/primary olfactory function, our research target mainly focused on the CYP genes that were highly expressed in the tentacles and low expressed or even not expressed in other tissues, so we thought that the "highly expressed" this expression would be better.
- Line 12 and 15. Include worker ant.
Response: We had corrected line 12 and 15 from “worker” to “worker ant”, thank you for your suggestion.
- Line 21 Change subfamilies by family.
Response: Thank you very much for your advice, also we had checked the naming categories of CYP and corrected the subfamilies in the text line 22-23 error to family.
- Include the objective of the research.
Response: Thank you very much for your suggestion, and we had added a description of the study aims in the abstract section (lines 19-20): "To verify whether the highly expressed CYPs in the antennae play an olfactory function in Solenopsis. invicta workers".
- Line 28. Workers with or without dsCYP?
Response: Based on your suggestion, we have corrected the original text from “Further, the trajectory behavior of the workers in response to EDMP revealed a significant reduction in both the range and speed of movement exhibited by the works in response to the chemical stimulus” (lines 29-31) to “Furthermore, the trajectory behavior assay showed that the worker's range and speed of movement in response to EDMP significantly decreased after the furthermore, the trajectory behavior assay showed that the worker's range and speed of movement in response to EDMP significantly decreased after the silencing of SinvCYP6K1 and SinvCYP4V2”.
- Line 44-45. This sentence has already been mentioned before, delete.
Response: Thank you for your suggestion, we had removed the duplicate sentence, you could see what we had changed in line 44.
- Line 53. Change ODE has by ODEs have…
Response: Thank you for your suggestion. We had corrected line 54 from “ODE has” to “ODEs have”".
- Line 54. Change It is by They are…
Response: As you suggested, we had changed line 55 from “It is” to “They are”.
- Line 56. Change It includes by ODEs include..
Response: As you suggested, we had changed line 57 from “It includes” to “ODEs include”.
- Line 59. Change heme superfamily by superfamily of hemo proteins
Response: As you suggested, we had changed line 60 from “heme superfamily” to “superfamily of hemo proteins”.
- Line 61. Include heme prosthetic group
Response: As you suggested, we had changed line 55 from “It is” to “They are”.
- Line 63. Include recognition of a wide range…
Response: Based on your suggestion, we had modified the original text (line 63-64)“The diversity of CYP enzymes and the wide range of substrates enable a multitude of distinct catalytic reactions” to “The diversity of CYP enzymes and include recognition of a wide range of substrates enable a multitude of distinct catalytic reactions”.
- Line 64-69 rewrite the paragraph
Response: In accordance with your request, the original paragraph had been modified to: “The enzymes present in the extract of the antennae of Phyllopertha diversa were observed to rapidly degrade a bioalkaline pheromone. The enzyme was found to be sensitive to CYP inhibitors and requires coenzyme NADPH activation, which indicated that CYP had a degrading function [14]. The EAG assay demonstrated that following the suppression of the antenna-specific P450 gene CYP4AW1 in P. diversa, the antenna's response to pheromones was significantly reduced, indicating that the gene is involved in the recognition process of pheromones [15].”
- Line 127. Include the names of the databases you used and the URL to them.
Response: It turned out that all of the sequence sources that were used in line 127 were in Table S5, and all of the URLs that were used had been added, as you could see in lines 133-141.
- Line 128. What do you mean by sequence numbers of the genes.
Response: The Gene Sequence Number is the accession number of the gene in the transcriptome based on the reference genome and allows a direct search for the gene on NCBI.
- Line 129. BLAST not defined when first mentioned
Response: Thank you for your suggestion, we have rewritten methodology2.4 to make it clearer. Below is the revised Section 2.4:
2.4 Multiple sequence comparison and phylogenetic analysis
Genes annotated as cytochrome P450 (CYP) were found through analysis of various databases in transcriptome data. Gene sequences were obtained from NCBI and then compared using Clustal W (https://www.genome.jp/tools-bin/clustalw) for multiple sequence comparison. Finally, the phylogenetic tree was constructed using the neighbour-joining method in MEGA5.2 software, and the values on the nodes indicate the percentage based on 1000 replicates.
- Line 131. Why neighbour-joining? I suggest use maximum likelihood method, include the access number of the sequences and species names used for phylogenetic reconstruction.
Response: Thanks to your suggestion, we refered to many similar studies in which the method used to construct the phylogenetic tree was the neighbor-joining method. The accession numbers and species names of all sequences used for phylogenetic reconstruction were listed in Table S5.
References:
Wu, H.; Liu, J.; Liu, Y.; Abbas, M.; Zhang, Y.; Kong, W.; Zhao, F.; Zhang, X.; Zhang, J. CYP6FD5, an antenna-specific P450 gene, is potentially involved in the host plant recognition in Locusta migratoria. Pestic. Biochem. Physiol. 2022, 188, 105-255.
Liu, F.; Xu, F.; Zhang, Y.; Qian, Y.; Zhang, G.; Shi, L.; Peng, L. Comparative Analyses of Reproductive Caste Types Reveal Vitellogenin Genes Involved in Queen Fertility in Solenopsis invicta. Int J Mol Sci. 2023, 24, 17130.
- Line 131. Missing URL of MEGA5.2
Response: There is no URL here because MEGA 5.2 is a computer local program.
- Line 133. Separate the physico-chemical characteristics in another section.
Response: We had taken your comments into account and we had a separate section for physico-chemical properties(line 122-132).
- Lines 134-138. Include this information in the expression of CYP’s section and add how to calculate of FPKM and differential expression?
Response: Thank you for your suggestion. Following your advice, we have added the information you mentioned to the section on the expression of CYPs and have revised the methods section accordingly. You can find our revisions in Method 2.3.
- Lines 140-144. What was the purpose of the cDNA synthesized from the section 2.1?
Response: We were grateful for your inquiry. The objective of the synthesized cDNA in Section 2.1 was to illustrate the particular process of cDNA synthesis. In response to your recommendation, we had revised the methodology of this section and incorporated the specific steps of the synthesized cDNA into Section 2.5. This modification would enhance the logical coherence of the article.
- Line 154-155. Eliminate this sentence.
Response: We were grateful for your proposal. “The candidate CYPs were screened based on tissue expression profiles and developmental expression profiles to be specifically highly expressed in the antennae of S. invicta”in question had been removed, and the amended text could be found in lines 165.
- Line 155-162. What was the purpose of doing an end-point PCR?
Response: The objective of the polymerase chain reaction (PCR) in this section was to obtain the correct target gene deoxyribonucleic acid (DNA), which would then be used as a template for subsequent specific double-stranded RNA (dsRNA) synthesis.
- Line 156. Insert the accession number of the genome.
Response: We were grateful for your proposal. The gene logins utilized in the experiment were accurately reflected in Result 3.1. However, to enhance the clarity of the methodology, we had also included the gene logins employed in Line 166. Please refer to the revision in Line 166.
- Line 164. Please define GFP.
Response: Green fluorescent protein (GFP) was frequently employed as a control in RNAi experiments, with the objective of demonstrating that the introduction of exogenous dsRNA into S. invicta did not affect the expression of the target gene.
- Line 172. Triplicate
Response: We would like to express my gratitude for your meticulous attention to the text. We had rectified an erroneous term and offer my sincere apologies for this inadvertent error. We were indebted to you for bringing this to my attention. Please refer to line 177 for details of the revision we had made.
- Line 176. Which components?
Response: In this context, the term “warning pheromone components” referred to the compound 2-ethyl-3,6(5)-dimethylpyrazine. To enhance clarity, we had revised this section of the paper (Line 187).
- Line 179. 20 mL or 20 uL?
Response: We would like to express our gratitude for your meticulous attention to detail. The correct volume is 20 μL, for which we were indebted to you for the timely reminder.
- Line 186-190. Please include what type of insects you use, ds-CYP fed or unfed. How many insects used?
Response: The behavioral responses of single-headed worker ants to 0.1 μg/μL, 1.0 μg/μL, and 10.0 μg/μL EDMP were recorded individually in each video. This was done after the ants had been fed specific dsRNA and control dsGFP. Three replicates were set for each concentration. Please refer to line 198-201 for details of the revision we had made.
- Line 191. Improve the description of statistical analyses.
Response: We were grateful for your valuable input, which had led to an enhancement of the description of the statistical analysis section. The methodology of this section was now more transparent and accessible. The changes to our methodology could be seen in line 199-205, where the paragraph had been revised to: “Data were presented as mean ± standard deviation (SD). Analysis of one-way variance (ANOVA) followed by Tukey’s HSD test (P < 0.05) was used to analyze the tissue and developmental stage expressions of SinvCYP gene, EAG responses and trajectory behavior. The silencing efficiency of dsSinvCYP and dsGFP were conducted using an Student’s t-test (*P < 0.05, **P < 0.01, ***P < 0.001). The data underwent analysis using the SPSS software (v. 23.0; SPSS Inc.), and the results of the statistical analyses were subsequently graphed using GraphPad Prism 9.0.”
- Line 202. Previously when??
Response: In accordance with your recommendation, the term "Previously" had been removed from Line 217.
- Lines 202-204. This is an interpretation of results, review this section and just describe the results.
Response: We are grateful for your valuable input and have taken it into consideration. In response, we have removed the description “The antennal CYPs may be involved in the recognition of odor and pheromone molecules, particularly those that are highly expressed in the antennae.” from the results section (line 217).
- Table 1. How was assignment the CYP names? And It is correct the length of CYP6K1-2?
Response: In the Results section, for example, the entries “CYP6K1, CYP6K1 -1, CYP6K1 -2” are all designated as CYP6K1 in NCBI. However, their gene sequences were not identical, comprising three distinct genes. To facilitate differentiation, the “-1, -2” suffixes were appended to distinguish between them. The designation “-1, -2” was used to distinguish between the two genes. Furthermore, an examination of the NCBI database revealed that CYP6K1-2 (Gene Bank: XM_026131197.2) was a protein comprising 1,064 amino acids in length.
- Line 218. Change Hlix K by K-Helix.
Response: In accordance with your recommendation, the erroneous term "Hlix" had been amended to “Helix” (line 237-238). Additionally, the expression "Helix K" had been duly referenced in the published paper:
Wu, H.; Liu, J.; Liu, Y.; Abbas, M.; Zhang, Y.; Kong, W.; Zhao, F.; Zhang, X.; Zhang, J. CYP6FD5, an antenna-specific P450 gene, is potentially involved in the host plant recognition in Locusta migratoria. Pestic. Biochem. Physiol. 2022, 188, 105-255.
- 3. Please indicate the units of expression
Response: The relative expression levels of the genes were obtained by calculation using the 2-ΔΔCT method, which indicated the change in expression level of the target gene relative to the housekeeper gene, and therefore had no labeled units.
Reference:
Lizana, P.; Mutis, A.; Palma-Millanao, R.; Larama, G.; Antony, B.; Quiroz, A.; Venthur, H. Transcriptomic and gene expression analysis of chemosensory genes from white grubs of Hylamorpha elegans (Coleoptera: Scarabaeidae), a subterranean pest in south America. Insects. 2024, 15(9): 660.
- Lines 277-279. Remove this paragraph.
Response: In accordance with your recommendation, “The results of RT-qPCR demonstrated that SinvCYP6K1 and SinvCYP4V2, exhibited markedly high expression levels in the antennae of worker ants. To further substantiate the role of these two genes in the recognition of alarm pheromones by S. invicta, we employed a feeding method to silent these two genes in S. invicta.” in question had been removed from the text.
- Line 282, Fig. 5 and 6. Please use h by hours.
Response: We were grateful for your valuable input and had revised the original text “hours” to “h”.
- Line 294. Different font size.
Response: We would like to express our gratitude for your meticulous attention to the text. We acknowledged that there was a discrepancy in the font formatting in line 307. We had taken the necessary steps to rectify the issue by standardizing the font.
- Line 301. Workers ants fed with dsCYP6K1…..
Response: We are grateful for your valuable input and have made the necessary revisions to the section “The feeding of” to read “Workers are fed with” (line 314).
- Line 304. Why these concentrations? Include this information in Material and Methods.
Response: The rationale behind the selection of these specific concentrations had been incorporated into the methodology section (line 188).
- Line 311. Eliminate letter T.
Response: I would like to express my gratitude for your meticulous attention to the text. Please note that the superfluous letter “T” has been eliminated (line 325).
- Discussion section. Rebuild this section.
Response: Thank you for your suggestion; we have rewritten the discussion section, and you can see our revisions in the manuscript.
- Line 340. This cite has a different format.
Response: In light of your invaluable feedback, the document had undergone a series of revisions.
Reviewer 3 Report
Comments and Suggestions for Authors
Jiang et al. investigate two antennal-specific cytochrome P450 genes of the red imported fire ant Solenopsis invicta, SinvCYP6K1 and SinvCYP4V2, for their potential role in olfactory recognition of alarm pheromones. Through transcriptomics, RT-qPCR, RNAi, and EAG bioassays, this work identified these two genes taking part in the detection of 2-ethyl-3,6(5)-dimethylpyrazine. Strong points: rich multi-method approaches besides sound statistical analysis. However, the flaws lie in inconsistent contextualization of findings within existing literature and shallow exploration of other roles for CYPs. More critical discussion needs to be made on functional validations, and deeper explorations on behavioral mechanisms are warranted. Language quality: 8/10. Overall rating: 83/100.
Major points:
This study demonstrated that EAG responses are reduced after RNAi silencing, and it was not possible to establish a direct enzymatic interaction between SinvCYPs and EDMP, leaving their exact functional role ambiguous within either pheromone degradation or signaling pathways.
Although DeepLabCut software allows one to glean insights into the innovation of behavioral dynamics, this analysis is severely hampered by a lack of supportive evidence linking the observed changes to altered olfactory perception at the neural level.
Exclusively depending on the EAG response as a proxy for olfactory recognition can be reductive; more neurophysiological data is necessary for the comparison.
The phylogenetic clustering of SinvCYP6K1 and SinvCYP4V2 with other insect CYPs, based on the indicated functional similarities, remains experimentally unverified for predicted functions in olfactory pathways, overextending the conclusion.
Where the statistical analysis is indeed appropriate and very robust, a graphical presentation would ideally plot group variations on key findings and test results concerning dose-dependent responses.
The RNAi methodology does not adequately address potential off-target effects, which could confound interpretations regarding gene function specificity.
Minor points
The introduction mentions a paucity of research on CYPs in alarm pheromone recognition but overlooks notable studies in related Hymenopteran species.
Figures lack uniformity in formatting and labeling, detracting from overall clarity (e.g., inconsistent scaling in bar charts).
The abstract overstates the novelty of findings without fully acknowledging limitations or alternative explanations for the observed phenomena.
References to previous studies in the discussion could be more explicit in distinguishing between corroborative and conflicting evidence.
Methodological details, such as RNAi feeding protocols, need further specificity to enhance reproducibility.
The supplementary materials are inadequately integrated into the main text, missing opportunities to substantiate claims with additional data.
The results have furnished important insights into the S. invicta alarm pheromone recognition mechanisms at the molecular level. These findings could find application in pest management via olfactory-based behavioral disruption, though they need further validation both in terms of molecular interactions and ecological consequences. The method is novel; however, it does not really lead to practical conclusions, as it needs supplementary mechanistic studies.
Author Response
Response to Reviewer 3 Comments
- Summary
Jiang et al. investigate two antennal-specific cytochrome P450 genes of Solenopsis invicta, SinvCYP6K1 and SinvCYP4V2, for their potential role in olfactory recognition of alarm pheromones. Through transcriptomics, RT-qPCR, RNAi, and EAG bioassays, this work identified these two genes taking part in the detection of 2-ethyl-3,6(5)-dimethylpyrazine. Strong points: rich multi-method approaches besides sound statistical analysis. However, the flaws lie in inconsistent contextualization of findings within existing literature and shallow exploration of other roles for CYPs. More critical discussion needs to be made on functional validations, and deeper explorations on behavioral mechanisms are warranted. Language quality: 8/10. Overall rating: 83/100.
Response: Thank the reviewer for your valuable feedback on our manuscript. We sincerely appreciate these positive remarks and for pointing out areas where further explanation is needed to enhance the clarity of our experiments. We are grateful for the opportunity to address these concerns before accepting the paper. We have carefully reviewed the comments and have made the necessary revisions accordingly.
Major Comments:
- This study demonstrated that EAG responses are reduced after RNAi silencing, and it was not possible to establish a direct enzymatic interaction between SinvCYPs and EDMP, leaving their exact functional role ambiguous within either pheromone degradation or signaling pathways.
Response: Inhibition of the expression of the odor-degrading enzyme genes might prevent the timely inactivation of odorants, which may lead to a decrease in the sensitivity of olfactory receptor neurons to odorants. This might result in a reduction in the insect's EAG response to odor stimuli (Wu et al., 2022; Feng et al. 2017; Maïbèche-Coisne et al. 2004). Accordingly, the present study employed the EAG to ascertain the alterations in the response values of red fire ant antennae to pheromones prior to and following the interference treatment. This approach might potentially offer insights into the involvement of CYPs in the olfactory recognition of pheromones.
References:
Maïbèche-Coisne, M.; Nikonov, A.A.; Ishida, Y.; Jacquin-Joly, E.; Leal, W.S. Pheromone anosmia in a scarab beetle induced by in vivo inhibition of a pheromone-degrading enzyme. Proc. Natl. Acad. Sci. U S A. 2004, 101, 11459-11464.
Feng, B.; Zheng, K.; Li, C.; Guo, Q.; Du, Y. A cytochrome P450 gene plays a role in the recognition of sex pheromones in the tobacco cutworm, Spodoptera litura. Insect Mol. Biol. 2017, 26, 369-381.
Wu, H.; Liu, J.; Liu, Y.; Abbas, M.; Zhang, Y.; Kong, W.; Zhao, F.; Zhang, X.; Zhang, J. CYP6FD5, an antenna-specific P450 gene, is potentially involved in the host plant recognition in Locusta migratoria. Pestic. Biochem. Physiol. 2022, 188, 105-255.
- Although DeepLabCut software allows one to glean insights into the innovation of behavioral dynamics, this analysis is severely hampered by a lack of supportive evidence linking the observed changes to altered olfactory perception at the neural level.
Response: We were grateful for your insightful remarks regarding this study. Utilizing DeepLabCut software enables observation of the reaction of S. invicta worker ants to the components of the warning pheromone subsequent to the specific CYP genes of worker ants being disrupted. The results of the study, obtained with DeepLabCut software, indicate that a reduction in the expression of the SinvCYP6K1 and SinvCYP4V2 in worker ants results in a decrease in both the speed and range of movement of the worker ants in response to the warning pheromone. Furthermore, the range of movement exhibited by the worker ants was observed to decrease when SinvCYP6K1 and SinvCYP4V2 were normally expressed. Conversely, when SinvCYP6K1 and SinvCYP4V2 expression was reduced, the movement speed and range of motion of the worker ants in response to the alarm pheromone were significantly higher. This evidence suggested that SinvCYP6K1 and SinvCYP4V2 may play a role in the olfactory recognition of the red fire ants by the components of the warning pheromone.
- Exclusively depending on the EAG response as a proxy for olfactory recognition can be reductive; more neurophysiological data is necessary for the comparison.
Response: The expression of genes that inhibit odor-degrading enzymes might affect the behavioral response of insects to odors (Chiu et al. 2019). In this study, a reduction in the expression level of single and double genes resulted in a corresponding decrease in the EAG response value, which in turn led to a notable decline in the behavioral response of worker ants to the alarm pheromone. This observation lend support to the notion that it was more prudent to hypothesize the olfactory role of CYPs based on EAG and insect behavior assays.
References:
Chiu, C.C.; Keeling, C.I.; Bohlmann, J. Cytochromes P450 preferentially expressed in antennae of the mountain pine beetle. J. Chem. Ecol. 2019, 45, 178–186.
- The phylogenetic clustering of SinvCYP6K1 and SinvCYP4V2 with other insect CYPs, based on the indicated functional similarities, remains experimentally unverified for predicted functions in olfactory pathways, overextending the conclusion.
Response: In light of your proposal, we had conducted a phylogenetic analysis of SinvCYP6K1 and SinvCYP4V2 in conjunction with other insect CYPs, given that protein structure was a determinant of function. Our hypothesis, therefore, was that SinvCYP6K1 and SinvCYP4V2 may perform analogous functions to other CYPs that have olfactory functions in other insects. The results of EAG and behavioral experiments indicated that both SinvCYP6K1 and SinvCYP4V2 were involved in the process of pheromone recognition by worker ants.
- Where the statistical analysis is indeed appropriate and very robust, a graphical presentation would ideally plot group variations on key findings and test results concerning dose-dependent responses.
Response: This figure represented the variability of results from dose-response experiments. We were grateful for your suggestion; indeed, graphical representation was a more suitable depiction of the pertinent dose effects. However, the objective of our study, as presented in this paper, was to demonstrate the EAG response of S. invicta worker ants to the warning pheromone following disruption of the target genes. The dose effect was not the primary focus of our investigation.
- The RNAi methodology does not adequately address potential off-target effects, which could confound interpretations regarding gene function specificity.
Response: Thank you very much for your suggestion. The off-target effects of RNAi are indeed an important issue that needs attention. In our experiments, we used specific RNAi sequences and also verified the primer specificity during the primer design process.
Minor Comments:
- The introduction mentions a paucity of research on CYPs in alarm pheromone recognition but overlooks notable studies in related Hymenopteran species.
Response: We were grateful for the observation that the "Nevertheless, the existing literature on the subject indicated that there is a paucity of research investigating the role of the CYP genes in the olfactory recognition of alarm pheromones" we discussed in the abstract, which encompasses the study of pheromones by CYP in Hymenopteran species, is more centered on its detoxification mechanism and less on the process involved in Hymenopteran olfactory function.
- Figures lack uniformity in formatting and labeling, detracting from overall clarity (e.g., inconsistent scaling in bar charts).
Response: In accordance with your recommendation, all images and bar charts within the text had been formatted in a consistent manner. Additionally, the scaling of bar charts with varying levels of consistency had been modified to ensure uniformity. The revisions were visible in the results section.
- The abstract overstates the novelty of findings without fully acknowledging limitations or alternative explanations for the observed phenomena.
Response: We were grateful for your proposal. The results of this study indicated that the SinvCYP6K1 and SinvCYP4V2 genes are involved in the olfactory function of S. invicta in response to the warning pheromone. The content mentioned in Lines 31-32 represented our interpretation of the experimental results presented in this paper. Additionally, we have removed the sentence "This provides a theoretical foundation for the potential development of olfactory-based S. invicta behavioral regulation technology" from Line 32-34 of the abstract. This provided a theoretical foundation for the potential development of olfactory-based S. invicta behavioral regulation technology in the abstract.
- References to previous studies in the discussion could be more explicit in distinguishing between corroborative and conflicting evidence.
Response: We were grateful for your valuable input and have incorporated it into our revised discussion section, which now included numerous citations to prior studies.
- Methodological details, such as RNAi feeding protocols, need further specificity to enhance reproducibility.
Response: We were grateful for your valuable input. In the Methods section, we had revised a portion of the methodology and incorporated additional details to enhance the replicability of the procedure.
- The supplementary materials are inadequately integrated into the main text, missing opportunities to substantiate claims with additional data.
Response: We were grateful for your valuable input. We had conducted a comprehensive review of the supplementary materials included in our article and can confirm that all relevant supplementary materials are duly referenced in our manuscript.
Round 2
Reviewer 1 Report
Comments and Suggestions for Authors
Thank you for responding to all comments. There is an error in text editing, since everything should have the same font size and there are a couple of scientific names that are not in italics (lines 89, 90, 94, 330, 424).
Line 107: transcriptome.
Author Response
Response to Reviewer 1 Comments
Dear reviewers:
Sincere thank you for the response received from the reviewers on our manuscript. We found them very helpful in improving the quality of manuscript. We have examined the comments and provide a point-by-point response.
Comments 1: Thank you for responding to all comments. There is an error in text editing, since everything should have the same font size and there are a couple of scientific names that are not in italics (lines 89, 90, 94, 330, 424).
Response 1: Thank you for your suggestion,we have reviewed and modified the font and scientific names throughout the manuscript.
Comments 2: Line 107: transcriptome.
Response 2: Thank you for your careful reading; we have corrected this error at line 119.
Reviewer 2 Report
Comments and Suggestions for Authors
Thank you for attending to my suggestions, however, there are still some details.
Abstract. Improve the sentence after including the objective of the article.
Line 12 and 20. Eliminate the dot after genera name.
Line 63 and 66 and the rest of the manuscript. Write in italics the species name.
Line 109. Which kind of RNA stabilizer do you use?
Line 135. I suggest use maximum likelihood method.
Line 144. cDNA was not analysed was sinthesised. I suggest use The cDNA was synthesised from 1 ug/uL of RNA using……
Line 161. Include the references of comparative Ct method
Author Response
Response to Reviewer 2 Comments
Dear reviewers:
Sincere thank you for the response received from the reviewers on our manuscript. We found them very helpful in improving the quality of manuscript. We have examined the comments and provide a point-by-point response.
Comments 1: Abstract. Improve the sentence after including the objective of the article.
Response 1: Thank you for your suggestion, We have refined the sentence in the abstract.
Comments 2: Line 12 and 20. Eliminate the dot after genera name.
Response 2: Thank you for your careful reading, We have already corrected this error in lines 12-21.
Comments 3: Line 63 and 66 and the rest of the manuscript. Write in italics the species name.
Response 3: Thank you for your suggestion. We have changed the species names to italics in line 76,81 and have checked the entire manuscript for species names to make the necessary corrections.
Comments 4: Line 109. Which kind of RNA stabilizer do you use?
Response 4: Thank you very much for your question. In this experiment, we used RNA later, and we have added the specific information about RNA later at line 121-122.
Comments 5: Line 135. I suggest use maximum likelihood method.
Response 5: Thank you for your suggestion. Following your advice, we have reconstructed the phylogenetic tree using the Maximum Likelihood method, and the modifications can be seen in both Method(2.4) and the Results(3.2).
Comments 6: Line 144. cDNA was not analysed was sinthesised. I suggest use The c DNA was synthesised from 1 ug/uL of RNA using
Response 6: Thank you for your suggestion, we have modified the text at line 159 to read "DNA was synthesised from 1 µg/µL of RNA using."
Comments 7: Line 161. Include the references of comparative Ct method
Response 7: Thank you for your suggestion, we have added the relevant reference at this point(line 174).